# Transcriptome Profiling of Developing Testes and First Wave of Spermatogenesis in the Rat

**DOI:** 10.3390/genes14010229

**Published:** 2023-01-16

**Authors:** Yan Zhang, Zaixia Liu, Xia Yun, Baiyin Batu, Zheng Yang, Xinlai Zhang, Wenguang Zhang, Taodi Liu

**Affiliations:** 1Medical Neurobiology Laboratory, School of Basic Medical Sciences, Inner Mongolia Medical University, Hohhot 010110, China; 2College of Animal Science, Inner Mongolia Agricultural University, Hohhot 010018, China

**Keywords:** differentially expressed genes, spermatogonial stem cells, spermatogenesis, transcriptome

## Abstract

Spermatogenesis is a complicated course of several rigorous restrained steps that spermatogonial stem cells undergo to develop into highly specialized spermatozoa; however, specific genes and signal pathways, which regulate the amplification, differentiation and maturation of these cells, remain unclear. We performed bioinformatics analyses to investigate the dynamic changes of the gene expression patterns at three time points in the course of the first wave of rat spermatogenesis. Differently expressed genes (DEGs) were identified, and the features of DEGs were further analyzed with GO (Gene Ontology), KEGG (Kyoto Encyclopedia of Genes and Genomes) and Short Time-series Expression Miner (STEM). A total of 2954 differentially expressed genes were identified. By using STEM, the top 10 key genes were selected in the profile according to the enrichment results, and the distinguishable biological functions encoded by these DEGs were automatically divided into three parts. Genes from 6, 8 and 10 days were related to biosynthesis, immune response and cell junction, and genes from 14, 15 and 16 days were related to energy metabolic pathways. The results also suggest that genes from 29, 31 and 35 days may shift metabolic to sperm motility, sperm flagellum and cilium movement.

## 1. Introduction

Male germ cells are a highly specialized type of cells, and they undergo significant morphological, biochemical and molecular changes during their development. Those cells start from the first germline cells in the embryo, Primordial germ cells (PGCs), and end with the production of haploid matured sperm [1]. In male rodent embryos, PGCs produce mitotically active germ cells, which later advance into a mitotic arrest in the mid-late fetal development period. Mitosis does not resume until a few days after birth, where then those cells take two different developmental fates. A number of the cells develop into spermatogonial stem cells (SSCs), which can self-renew and differentiate, while the others directly differentiate without self-renewal, making the first wave of spermatogenesis occur [2]. Spermatogenesis provides male mammals with a constant supply of sperm throughout their lives and ensures that genetic information is passed from one generation to the next [1]. The rodent spermatogenic cycle starts with mitotic and, through mitosis, produces many type A and type B spermatogonia. Type B spermatogonial cells enlarge to take the shape of primary spermatocytes, then meiotic begins. Secondary spermatocytes and round spermatids are generated after meiosis Ⅰ and Ⅱ. Lastly, mature sperm are formed from spermatids via spermiogenesis, during which significant cellular morphological and structural changes take place, including condensation of chromatin, shaping of nuclear, removal of extra cytoplasm, and formation of the sperm tail and acrosome [3]. The process of spermatogenesis is similar in different species, but the time it takes is different, with approximately 35 days in mice [4] and 50 days in rats [5].

The rat testis contains hundreds of seminiferous tubules, which include germ cells at different developmental periods during spermatogenesis. In order to characterize the changes during spermatogenesis, we should determine all the time points when those changes are occurring. Timing is critical for tissue development in transcriptomic studies, and luckily, the timeline for the first wave of spermatogenesis in rats has been established [6]. Throughout the first wave of spermatogenesis, the critical time point at which spermatogenic cells appear is now well-defined. At postnatal day (PND) 9, seminiferous tubules contain only spermatogonia and Sertoli cells (SCs). Primary spermatocytes emerge at PND 15, pachytene spermatocytes are first observed at PND 18, and diplotene spermatocytes are observed at PND 24. After meiosis, round spermatids emerge at PND 26 and elongating spermatids appear at around PND 33–38, and at PND 42, elongated spermatids are formed [5]. This phase of germ cell development is, therefore, crucial to initiate and maintain male fertility throughout adult life. Rodents of different ages could be sacrificed, and seminiferous tubules of specific spermatogenic stages could be collected, which may have unique patterns of germ cell composition [6]. After collecting rat testicular tissues of different ages, RNA could be extracted for sequencing.

The mRNA population specifies a cell’s identity and helps to govern its present and future activities [7]. RNA sequencing technology (RNA-Seq) offers a universal view of gene expression profiles; this technology has helped to develop the mapping and quantifying of the transcriptome [8]. This technique has various superiority over other transcriptome methodologies, for example, higher resolving power and sensibility, a large dynamic range of gene expression levels, the ability to identify original transcribed regions and splicing isoforms of known genes [8]. Gene expression during spermatogenesis is extremely orchestrated and tightly controlled at the transcriptional and post-transcriptional levels. Several specific transcriptional regulators are required to complete the complicated development of spermatozoa [9]. It is reported that approximately 30% of the genome is differentially expressed during murine testis development. The expressional differences in rat testis mainly occur in the first few days after birth (PND 6–10), in the initial stage of meiosis (PND 14–16), and with the emergence of haploid sperm (around PND 30) [10]. RNA-Seq could qualify the global examination of gene expression of transcript content in the sample and could distinguish and quantify rare transcripts and supply information concerning alternative splicing and sequence variation in genes that have been identified.

Here, the differences in gene expression during the first wave of spermatogenesis in rats were analyzed. Total testicular tissue samples were collected at three critical time points: at PND 6, 8 and 10, PND 14, 15 and 16, and at PND 29, 31 and 35. Currently, almost all the studies about testicular development and spermatogenesis are concentrated on mice or humans. The complicated physiology of the rat reflects many human traits, which could provide the ability to delve into the potential pathways and regulatory mechanisms of polygenic diseases. This study provides a very valuable basis for studying the genetic genes of male fertility and their regulatory and influencing pathways.

## 2. Materials and Methods

### 2.1. Animals and Sample Collection

Wistar-Imamichi rats were placed in 12 h of light/12 h of darkness and allowed autonomous intake of water and food. Male and female rats naturally mate. Day 1 means the date of the rat’s birth. Rat pups were separately killed by cervical dislocation method on PND 6, 8, 10, 14, 15, 16, 29, 31 and 35, and testes were collected. At least three testes from different litters were collected at each time point. All procedures were performed in accordance with the NIH Guide for the Care and Use of Laboratory Animals.

Whole testes from rats of each time point were collected and snap-frozen in liquid nitrogen and stored at −80 °C prior to RNA extraction. The three phases of the measured microarrays were based on each stage of spermatogenesis in rats. The samples were sequenced by KangChen Bio-technology Co, Shanghai, China. A total of 27 samples of testes were used in the study.

Samples were collected at different time points in the course of the first wave of rat spermatogenesis. Each time point represents a different stage of development in the testis. PNDs 6, 8 and 10 are the first time points when most SSCs are present in the seminiferous tubules, and S was used to represent this time point. The second time point was selected at PNDs 14, 15 and 16 when meiosis began, and M was used to represent this time point. The last time point was at PNDs 29, 31 and 35, at which time the testis contained round and elongating spermatids, and R was used to represent this time point.

### 2.2. Microarray Analysis

Microarray experiments were conducted by KangChen Bio-technology Co. (Shanghai, China) according to the standard procedure [11]. Briefly, total RNAs from testes were extracted using Trizol (Invitrogen, Carlsbad, CA, USA) and RNeasy Mini Kit (Qiagen, Valencia, CA, USA) following the manufacturers’ instructions. Genomic DNA contamination was removed by DNase treatment. The concentration of RNA was measured through a Nanodrop ND-1000 Spectrophotometer (NanoDrop products, Wilmington, DE, USA). Precisely 10 mg of total RNA was taken to synthesize double-stranded cDNA, following the steps of an Invitrogen Superscript Double-Stranded cDNA Synthesis Kit (Invitrogen). RNA contamination was removed by RNase A, and cDNA was precipitated and labelled using a NimbleGen One-Color DNA Labeling Kit (RocheNimbleGen, Inc., Pleasanton, CA, USA). Labelled cDNA was quantitated and hybridized with microarray chip slides in the NimbleGen Hybridization System (Roche-NimbleGen, Inc.). Following hybridization and washing, the microarray slides were scanned with an Axon GenePix 4000B microarray scanner (Molecular Devices, Sunnyvale, CA, USA).

### 2.3. Differential Expression Analysis

For three periods, we identified the genes that are differentially expressed (DEGs) between two periods using the formula and limma package for the R language. For M vs. SSCs and R vs. SSCs, firstly, the gene intensity value was standardized, where xi is the gene intensity value, xj is the average gene intensity value of each sample, and stdxj is the standard deviation of the gene intensity value of each sample. There were 12,215 genes shared by SSCs and M, and 12,122 genes shared by SSCs and R. Secondly, FDR was calculated to correct the *p*-value, and the normalized gene intensity values were calculated as follows: FoldChange = ABS((average M/average SSCs) [12,13,14]. Finally, the genes were considered DEGs when |log2 FoldChange| > 3, FDR < 1 and *p*-value < 0.05. For R vs. M using the limma package, we required a *p*-value < 0.05 and |log2 FoldChange| > 3.

### 2.4. GO and KEGG Enrichment Analyses of DEGs

The DEGs functional enrichment analyses were conducted using the Metascape [15]. Three main ontologies, including biological process, molecular function and cellular component, were annotated for DEGs. GO terms and pathways of DEGs with a *p*-value < 0.01 were considered significantly enriched.

### 2.5. Series Cluster and Key Regulatory Genes Analysis

Series cluster analysis was performed to determine DEG expression profiles in the STEM program (https://www.cs.cmu.edu/~jernst/st/; accessed on 10 December 2020). The union of DEGs in M vs. SSCs, R vs. SSCs and R vs. M testes was categorized into five profiles. Subsequently, the Pearson correlation coefficient between genes in each profile was calculated to screen out R > 0.85 and *p*-value < 0.01, and the top ten key genes in the profile were selected according to the enrichment results, which may play a crucial role in gene interaction and regulation of spermatogenesis.

## 3. Results

### 3.1. Data Evaluation

The number of probes detected in the S, M and R stages was 19,683, 25,795 and 26,177, and the corresponding number of unique genes were 17,492, 17,269 and 17,359, respectively. The number of specifically expressed genes in the S, M and R stages was 4912, 744 and 926, respectively. The boxplot is a traditional method for visualizing the distribution of a dataset. Here, the boxplot view was used to view and compare the normalized distribution of sample expression values or experimental conditions, and it was found that the expression quantity was consistent in each period (Figure 1).

### 3.2. DEGs at Different Stages

DEGs at different stages were analyzed. The greatest number of DEGs were identified between the M vs. R stages. A total of 953 genes were up-regulated, and 73 genes were down-regulated. Specifically, 627 genes were up-regulated and 713 genes were down-regulated from the S to M stage, and 689 genes were up-regulated and 290 genes were down-regulated at the S stage compared with the R stage (Figure 2A). A greater number of DEGs were detected between the spermatid’s formation stages than in previous developmental stages. We identified that 390 genes were specifically up-regulated and 601 genes were down-regulated between the S vs. M stages, 430 genes were specifically up-regulated and 193 genes were down-regulated between the S vs. R stages, and 898 genes were specifically up-regulated and 63 genes were down-regulated between the R vs. M stages (Figure 2B).

### 3.3. Gene Ontology (GO) and KEGG Enrichment Analysis of DEGs

This study also employed a Metascape analysis method based on GO and KEGG gene sets to discover potential signaling pathways in spermatogenesis at different stages. We performed a GO classification and functional enrichment on DEGs of different stages in testis development. GO terms were assigned to DEGs from the S vs. M, S vs. R and M vs. R stage comparison. The GO analysis showed transcripts related to germ cell division processes, such as cell cycle process (GO: 0022402), chromosome organization (GO: 0051276), intracellular protein-containing complex (GO: 0140535), transferase complex (GO: 1990234), cell division (GO: 0051301) and response to oxidative stress (GO: 0006979) in both the S vs. M and S vs. R stages. However, DEGs between M and S were enriched in the regulation of the protein catabolic process (G0: 0042176) and Alzheimer’s disease (rno: 05010). Furthermore, GO analysis between R vs. M revealed genes related to spermiogenesis and cytoskeleton, such as epithelial cilium movement involved in the extracellular fluid movement (GO: 0003351), microtubule organizing center (GO: 0005815), sperm principal piece (GO: 0097228), axonemal dynein complex assembly (GO: 0070286), acrosome assembly (GO: 0001675), sperm connecting piece (GO: 0097224), microtubule-associated complex (GO: 0005875), manchette (GO: 0002177), regulation of cilium movement (GO: 0003352), cilium (GO: 0005929), spermatogenesis (GO: 0007283) and acrosomal vesicle (GO: 0001669) (Figure 3A).

Kyoto Encyclopedia of Genes and Genomes (KEGG) pathway enrichment analysis was performed on these differentially expressed genes with a cut-off criterion of corrected *p*-values < 0.01. The heat map is complemented by an enrichment network in which each network node is represented by a pie chart, where the sector size is proportional to the number of genes originating from each gene list. This representation is designed as a visual illustration of the biology of selective or shared pathway clusters (Figure 3B).

### 3.4. Screening and Analysis of Key Genes Using STEM

The gene regulatory network is a continuous and complex dynamic system, and the regulation of genes is a dynamic event that changes with time and the environment. The gene expression data are obtained by measuring the cell cycle and gene expression under specific state conditions at different time points; that is, time series gene expression data contain rich gene regulation information. If we want to look for genes of a particular type of expression, such as genes whose expression has been increased/decreased over time or increased first and then decreased, it is a Short Time-series Expression Miner (STEM). In series cluster analysis, we categorized the 2954 DEGs that were differentially expressed in three stages into five profiles (Figure 4). The number of genes grouped in each profile is shown at the bottom left corner of each square. The Pearson correlation coefficient of the genes in each profile was calculated, and the screen criteria were R > 0.85 and a *p*-value < 0.01. The top 10 key genes in the profile were selected according to the enrichment results, and these genes may play a key role in gene interaction and regulation. Short-time sequence expression analysis of DEGs was performed to understand the change of gene expression trend in the range of dynamic time series. STEM analysis of the 2954 DEGs showed that there were five expression patterns in three stages of spermatogenesis, among which 953 DEGs in module 49 showed an increased expression trend in SSCs, M and R, while 894 DEGs in module 0 showed a decreasing expression trend. In module 1, 326 DEGs showed a trend of first rising and then downloading expressions. The S-stage cells are enriched in certain pathways on an extracellular exosome, and RNA and cadherin binding are involved in cell–cell adhesion, DNA-directed RNA polymerase activity, and so on (Figure 5, Profiles 0 and 22). Obviously, M-stage cells are enriched in a metabolic pathway, nucleus mitochondrion and endosome MAPK cascade pathways (Figure 5, Profile 46). Pathways on sperm motility, motile cilium, sperm flagellum, cilium movement etc. are enriched in R-stage cells (Figure 5, Profiles 1 and 49).

## 4. Discussion

Spermatogenesis takes place in the wall of the seminiferous tubules and involves a multitude of sequential cell types. First, there are undifferentiated spermatogonia, among which lies a population of SSCs. SSCs can both self-renew and give rise to large numbers of spermatogonia [16]. Second, the undifferentiated spermatogonia, which are diploid, then undergo differentiation and further rounds of division that give rise to spermatocytes, which enter the meiotic process. During the lengthy meiotic prophase, recombination takes place through two sequential divisions. Parental and maternal chromosomes are randomly distributed into haploid daughter cells, which are termed spermatids [17]. Finally, the spermatids develop from round cells into highly specialized spermatozoa that are ultimately released into the seminiferous tubule lumen [18]. Analysis of cells collected directly from the physiological environment could eliminate anthropogenic changes in expression levels that may result from preparation and culture procedures. Here, we collected testicular samples from these three time points and identified a number of key genes and pathways involved in the first wave of sperm production.

Recently, there have been some studies on transcriptome profiling of the testis, but most of them are performed in mice. Up to now, almost every new drug has been tested on rats, and they have made great contributions to the pharmaceutical industry for decades. If we could understand the biology of the rat much better, then we could build perfect models of human disease which can be used to develop new drugs. Laiho A et al. showed that genes involved in gametogenesis, germ cell development, fertilization and sperm motility were up-regulated during the first wave of mouse spermatogenesis [9]. This matches well with the obtained from our current study. In our study, a total of 2945 DEGs were identified in the comparisons across successive time points, where some genes were up-regulated, and some were down-regulated. Functional analysis of DE genes disclosed enriched GO terms are related to spermatogenesis-associated procedures, such as meiosis, cell–cell adhesion, nucleus membrane and sperm motility. In addition, the specific genomes of each GO term showed up-regulation results at different points in time, highlighting the characteristics of each cellular process of spermatogenesis. Cell cycle process, chromosome organization, intracellular protein-containing complex and cell division were up-regulated during the transition from spermatogonia to early meiotic cell (S to M) and S to R, but regulation of protein catabolic process and Alzheimer’s disease-related genes were only enriched in S to M. Epithelial cilium movement included in the extracellular fluid flow, microtubule organizing center, sperm principal piece, axonemal dynein complex assembly and acrosome assembly-related terms were up-regulated during postmeiotic spermiogenesis (M to R).

The study by Guo et al. provided valuable information regarding gene expression dynamics during human spermatogenesis [19]. Their study indicated that the top five up-regulated GO terms were nuclear lumen, cell cycle/division, chromosome, endoplasmic reticulum and mRNA processing/splicing-related process, which is consistent with our results of SSCs stage cells (Figure 3, Profiles 0 and 22). Their top five up-regulated GO terms of early primary spermatocyte were ribosome/translation, cytoskeleton, spermatogenesis, mitochondrial and protein folding, which is similar to our meiosis stage cells (Figure 3, Profile 46). Their top five up-regulated GO terms of sperm were ribosome, spermatogenesis, ubiquitin and sperm motility, which is similar to our round spermatid stage cells (Figure 3, Profiles 1 and 49). Both studies indicated that the specificity of gene expression during spermatogenesis is a universal phenomenon in different species. Although there are differences in the time taken for a spermatogenic cycle to be completed between humans and rodents, the process itself is similar between the two species. Therefore, rodents are useful animal models for studying the spermatogenic process [1].

The first wave of spermatogenesis, bypassing the SSC step, was likely selected over evolutionary time because it permits the rapid generation of sperm, thereby bestowing fertility to young males [20]. Based on this study, we have identified a wealth of tissue-specific gene isoforms, which is crucial for the comprehension of the complexity of transcriptomes and gene regulation.

## 5. Conclusions

Our data elucidate the diversity of expressed genes and signal pathways during the first wave of rat spermatogenesis and lays the ground for further studies of specific factors contributing to male fertility. The number of probes detected in the S, M and R stages was 19,683, 25,795 and 26,177, and the corresponding number of unique genes was 17,492, 17,269 and 17,359, respectively. The number of specifically expressed genes in the S, M and R stages was 4912, 744 and 926, respectively. Most genes have functions within biological process categories and are known to participate in spermatogenesis and testis development. These findings will help us to understand further how gene expression is regulated during spermatogenesis. In future studies, it will be critical to perform decisive experiments to investigate those transcription factors.

## Figures and Tables

**Figure 1 genes-14-00229-f001:**
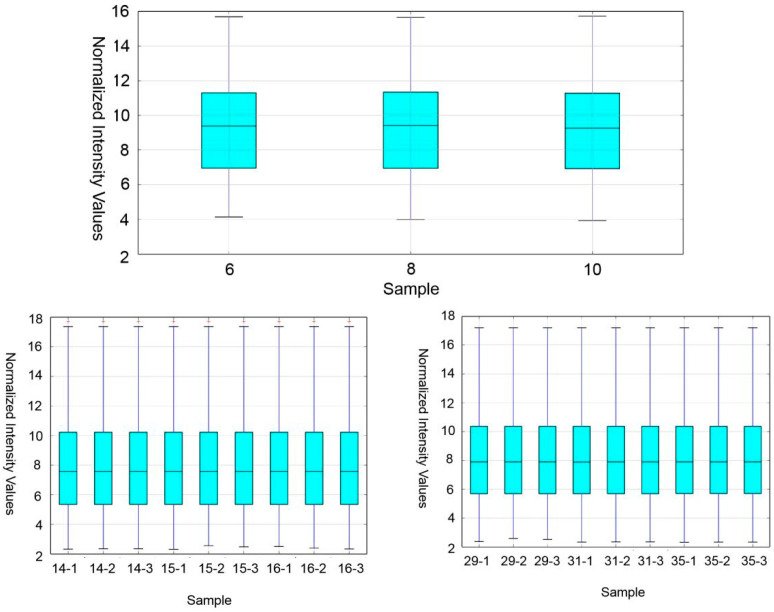
Box plot of normalized intensity values in three phases of spermatogenesis.

**Figure 2 genes-14-00229-f002:**
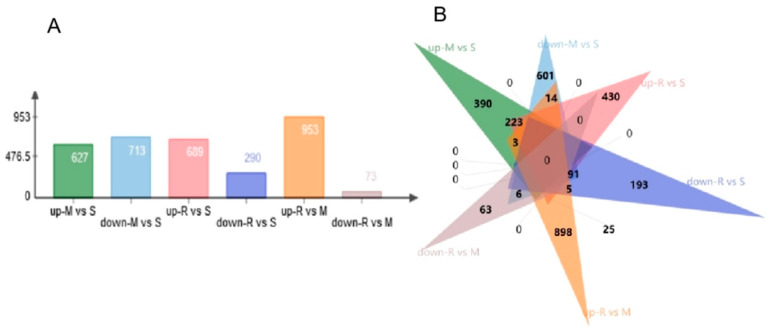
DEGs between the different testis developmental stages. (**A**) Summary of DEGs. The x-axis represents compared samples. The y-axis represents DEG numbers. (**B**) The Jvenn diagram of the number of differential up- and down-regulated genes in different reproductive phases.

**Figure 3 genes-14-00229-f003:**
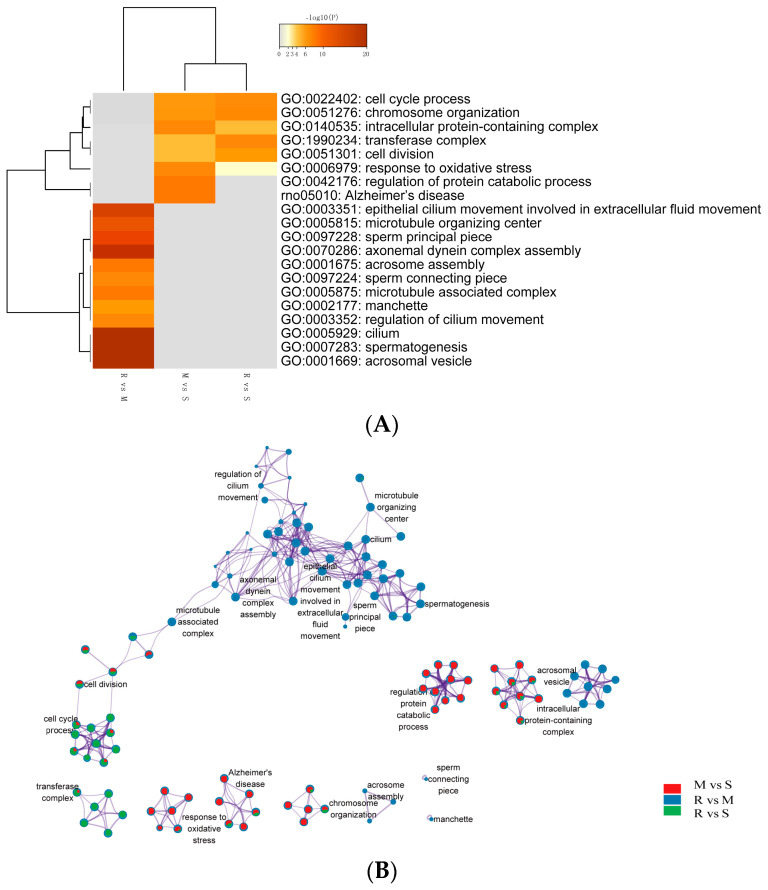
Visualizations of meta-analysis results based on multiple gene lists. (**A**) A heatmap showing the top enrichment clusters, one row per cluster, using a discrete color scale to represent statistical significance; gray color indicates a lack of significance. The categories GO:0022402 (cell cycle process) to GO: 0006979 (response to oxidative stress) are related to M vs. S and R vs. S. While GO:0046176 (regulation of protein catabolic process) and rno: 05010 (Alzheimer’s disease) are enriched exclusively in the M vs. S stage. From GO: 0003351 (epithelial cilium movement involved in extracellular fluid movement) to GO: 0001669 (acrosomal vesicle) are enriched exclusively in the R vs. M stage. (**B**) Enrichment network visualization for results from the three gene lists, where nodes are represented by pie charts indicating their associations with each input study. Cluster labels were manually added. Color code represents the identities of gene lists. The network shows that processes, such as cell division, cell cycle process and chromosome organization, are generally shared among the M vs. S and R vs. S stages. Process regulation of cilium movement and microtubule organizing center and so on are especially enriched between M vs. R (red represents the gene lists between the M vs. S stage; blue represents the M vs. R stage; green represents the R vs. S stage).

**Figure 4 genes-14-00229-f004:**
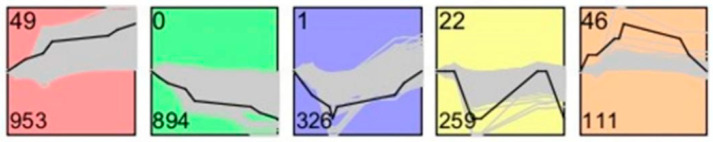
Five profiles of DEGs with unique expression alterations over SSCs, M and R. The profile number is shown at the top left corner of each square. The number of genes grouped in each profile is shown at the bottom left corner of each square. Profiles are ordered based on the number of genes assigned.

**Figure 5 genes-14-00229-f005:**
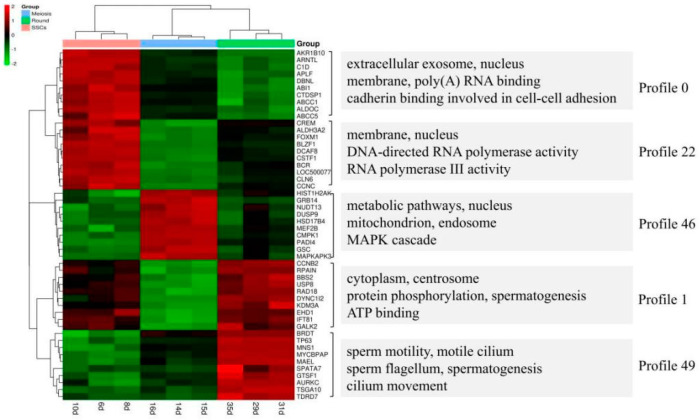
RNA-seq heat map of all three stages of cells with key genes and enriched GO terms at the right.

## Data Availability

Data is unavailable due to privacy or ethical restrictions.

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
