# Peer review of "Transcriptome Profiling of Developing Testes and First Wave of Spermatogenesis in the Rat"

_genes, 2023, doi:10.3390/genes14010229_

Round 1

Reviewer 1 Report

In this study, the authors analysed developing rat testes at the transcriptional level, at three different time periods during the first wave of spermatogenesis: i) at 6, 8, 10 days, e.g. before meiotic entry (“S stage”), ii) at 14, 15, 16 days, e.g. during meiosis (“M stage”) and iii) at 29, 31, 35 days, e.g. during spermiogenesis (“R stage”). A total of 2954 differentially expressed genes (DEGs) were identified. GO (Gene Ontology), KEGG (Kyoto Encyclopedia of Genes and Genomes) and Short Time-series Expression Miner (STEM) analyses were also performed. The authors found that genes involved in biosynthesis, immune response and cell junction were enriched at the S stage, genes involved in energy metabolism were enriched at the M stage and that genes involved in sperm motility were enriched at the R stage.

Although the topic is interesting, this study is only descriptive and biological validations of the data are lacking. The entire manuscript needs to be edited for English language, grammar, punctuation and spelling by an English-speaking person. Other modifications are also required in this manuscript.

-   Line 46: “42-45 days in rat”. In other publications, the duration of rat spermatogenesis is 48-53 days.

-   Line 127: “the gene intensity value was standardized using , where xi …”. Please clarify this sentence.

-   Line 131: “as follows: and ; ,”. Please clarify this sentence.

-   Fig 1: Please rotate the legend on the y axis

-   Fig 1: why were three samples not analyzed at 6, 8 and 10 days?

-   Fig2A : why showing this figure since the number of up-regulated and down-regulated genes between the different stages are already detailed in the text?

-   Line 170: Please replace 190 by 193

-   Line 212: Please replace “R vs R stage” by “R vs M stage”

-   Line 215-218: “The network shows that processes such as cell division, cell cycle process and chromosome organization are generally shared among M vs S and R vs S; Process regulation of cilium movement and microtubule organizing center and so on are enriched specially between M vs R.”

Why showing figure 3B since all these information are already presented in Figure 3A?

-   Please better explain Figure 4.

-   Overall, did the authors obtain novel findings or do these transcriptomic data only confirm the results already obtained in other species? In other words, what is the originality of this study?

-   The discussion needs to be rewritten as it looks more like an introduction and a repetition of the results section.

Author Response

-   Line 46: “42-45 days in rat”. In other publications, the duration of rat spermatogenesis is 48-53 days.

Reply:Thank you for reminding us. After consulting and comparing with classical papers, we accept your suggestion and modify the number of days to 50 days. The classical peaper is as follows, and we changed ref.5 into this one, too.

  1. Adler ID. Comparison of the duration of spermatogenesis between male rodents and humans. Mutat Res. 1996. 352(1-2):169-72.

-   Line 127: “the gene intensity value was standardized using , where xi …”. Please clarify this sentence.

Reply: The method for finding DEGs from lines 127 to 132 was a reference to these three papers and was optimized to ensure that the DEGs found had no false positives:

  1. Gou Y, Zheng X, Li W, et al. Polysaccharides Produced by the Mushroom Trametes robiniophila Murr Boosts the Sensitivity of Hepatoma Cells to Oxaliplatin via the miR-224-5p/ABCB1/P-gp Axis[J]. Integrative Cancer Therapies, 2022, 21: 15347354221090221.
  2. Besecker B, Bao S, Bohacova B, et al. The human zinc transporter SLC39A8 (Zip8) is critical in zinc-mediated cytoprotection in lung epithelia[J]. American Journal of Physiology-Lung Cellular and Molecular Physiology, 2008, 294(6): L1127-L1136.
  3. Galon Y, Aloni R, Nachmias D, et al. Calmodulin-binding transcription activator 1 mediates auxin signaling and responds to stresses in Arabidopsis[J]. Planta, 2010, 232(1): 165-178.

-   Line 131: “as follows: and ; ,”. Please clarify this sentence.

Reply: answer as above.

-   Fig 1: Please rotate the legend on the y axis

Reply: I'm very sorry that I didn't quite understand your meaning. What is the meaning of  “ rotate the legend on the y axis”. Here, Y axis represent that: Normalized Intensity of each sample, the gene expression summary value for the gene.

- Fig 1: why were three samples not analyzed at 6, 8 and 10 days?

Reply: This part of samples was the first to be sent for sequencing. At that time, due to the small volume of each sample, three samples were mixed and sequenced, so only one set of data was output.

- Fig2A : why showing this figure since the number of up-regulated and down-regulated genes between the different stages are already detailed in the text?

Reply: The figure shows the number of DEGs more directly, and the combination of Figure2 A and Figure2 B makes it more clear.

-   Line 170: Please replace 190 by 193

Reply: It has been replaced.Please see the red font in the revised version for details.

-   Line 212: Please replace “R vs R stage” by “R vs M stage”

Reply: It has been replaced. Please see the red font in the revised version for details.

-   Line 215-218: “The network shows that processes such as cell division, cell cycle process and chromosome organization are generally shared among M vs S and R vs S; Process regulation of cilium movement and microtubule organizing center and so on are enriched specially between M vs R.”

Why showing figure 3B since all these information are already presented in Figure 3A?

Reply: Figure 3B is added to highlight the interactions between the enriched terms. Most physiological functions do not exist independently and require multiple synergies.

-   Please better explain Figure 4.

Reply: Modified, please see line 241-247, the red font section.

-   Overall, did the authors obtain novel findings or do these transcriptomic data only confirm the results already obtained in other species? In other words, what is the originality of this study?

Reply:There are three reasons:

  • The first wave of spermatogenesisprovides an invaluable tool for characterization of gene expression and cellular events during spermatogenesis. But most of the research so far has focused on mice, and the first wave of spermatogenesis in rats has rarely been reported.
  • Analyzing cells at their physiological environment eliminates the possible changes in expressionlevel due to culture
  • Although a number of studies have beenable to elucidate the stage specificity and complexity of testicular transcription machinery, the overall picture of the gene expression patterns and transcript variety during spermatogenesis is still largely unknown. Our present work also aims to reveal the signaling pathways and key genes in spermatogenesis, and provide important theoretical basis and research direction for our following research work.

-   The discussion needs to be rewritten as it looks more like an introduction and a repetition of the results section.

Reply: Rewritten, please see the red font in discussion section in the revised draft.

Reviewer 2 Report

I checked your manuscript and described comments below.

This paper provides a very good bioinformatics analysis of rat spermatogenesis.

Especially the part "genes from 29, 31, 35 days may shift metabolic to sperm motility, sperm flagellum and cilium movement." is important.

I don't think this paper has any major mistakes or grammatical problems.

Author Response

Thank you very much for your affirmation of our scientific research work.

Reviewer 3 Report

This is an original paper focused on transcriptome profiling of developing testes and first wave of spermatogenesis in the rat and brings some new knowledge.  However, I also have a few reservations about the manuscript, which are listed below.

 Comments:

Section Animals and sample collection:

-          how were stored testes -  please add this information

-          what RNA sample protector was use - please add this information

-          it is not clear how the 9 samples were selected for analysis; authors present that at least three testes were collected at every time-point - each PND (6, 8, 10, 14, 15, 16, 29, 31 and 35) is considered as a time point in this case (if yes, was a mixed sample taken from multiple collections on the same day used or was a random sample selected), or S, M and R are considered as time points - please specify this part

 Section Differential expression analysis:

-          line 127 - missing model how gene intensity value was standardized

-          line 131 - missing model how normalized gene intensity values are calculated

 Section Conclusion

-  line 310 - “konwn” should be “known”.

Has quantitative real time PCR been done to validate DEGs? If so, which genes were selected for validation? -  please add the following information to the Methods and Results.

Author Response

Section Animals and sample collection:

- how were stored testes -  please add this information

Reply: Thanks for your reminding, this part has been added, please see the the red font in line101-102 in the revised version for details.

- what RNA sample protector was use - please add this information

Reply: Since the sequencing of this study was commissioned by a reagent company (Kangchen Bio-tech, shanghai, China). In order to obtain more prepared results, we directly mailed frozen testicular samples to the company, and they extracted RNA for direct sequencing. So we didn't use RNA protectors.

- it is not clear how the 9 samples were selected for analysis; authors present that at least three testes were collected at every time point - each PND (6, 8, 10, 14, 15, 16, 29, 31 and 35) is considered as a time point in this case (if yes, was a mixed sample taken from multiple collections on the same day used or was a random sample selected), or S, M and R are considered as time points, please specify this part 

Reply: I'm really sorry that we didn't write this part clearly in the paper. The 9 samples were collected at different points in time, divided into three periods, and each period was selected on three different days. There were three more replicates for each type of day (rats in different litters). Each testicle is individually labeled and mailed to the sequencing company. 

Section Differential expression analysis:

- line 127 - missing model how gene intensity value was standardized

- line 131 - missing model how normalized gene intensity values are calculated

Reply: The method for finding DEGs from lines 127 to 132 was a reference to these three papers and was optimized to ensure that the DEGs found had no false positives:

  1. Gou Y, Zheng X, Li W, et al. Polysaccharides Produced by the Mushroom Trametes robiniophila Murr Boosts the Sensitivity of Hepatoma Cells to Oxaliplatin via the miR-224-5p/ABCB1/P-gp Axis[J]. Integrative Cancer Therapies, 2022, 21: 15347354221090221.
  2. Besecker B, Bao S, Bohacova B, et al. The human zinc transporter SLC39A8 (Zip8) is critical in zinc-mediated cytoprotection in lung epithelia[J]. American Journal of Physiology-Lung Cellular and Molecular Physiology, 2008, 294(6): L1127-L1136.
  3. Galon Y, Aloni R, Nachmias D, et al. Calmodulin-binding transcription activator 1 mediates auxin signaling and responds to stresses in Arabidopsis[J]. Planta, 2010, 232(1): 165-178.

-  line 310 - “konwn” should be “known”.

Reply: It has been replaced. Please see the red font in the revised version for details.

Has quantitative real time PCR been done to validate DEGs? If so, which genes were selected for validation? -  please add the following information to the Methods and Results.

Reply: I am very sorry that we did not carry out this part of the experiment, we will focus on this part in the following research.

Round 2

Reviewer 3 Report

After reading the responses to the comments, I accept the explanations and corrections made by the authors.

I have one new comment:

If the method for finding degrees was implemented according to the following references (1. Gou Y, Zheng X, Li W, et al. Polysaccharides Produced by the Mushroom Trametes robiniophila Murr Boosts the Sensitivity of Hepatoma Cells to Oxaliplatin via the miR-224-5p/ABCB1/P-gp Axis[J]. Integrative Cancer Therapies, 2022, 21: 15347354221090221.; 2. Besecker B, Bao S, Bohacova B, et al. The human zinc transporter SLC39A8 (Zip8) is critical in zinc-mediated cytoprotection in lung epithelia[J]. American Journal of Physiology-Lung Cellular and Molecular Physiology, 2008, 294(6): L1127-L1136.; 3. Galon Y, Aloni R, Nachmias D, et al. Calmodulin-binding transcription activator 1 mediates auxin signaling and responds to stresses in Arabidopsis[J]. Planta, 2010, 232(1): 165-178.), add this information to the manuscript.

Author Response

We appreciate for your warm work earnestly. We have added the references to the text, please see line 138 in the revised version, and the three references are 12, 13 and 14 respectively .
